# Hexavalent Chromium Disrupts Oocyte Development in Rats by Elevating Oxidative Stress, DNA Double-Strand Breaks, Microtubule Disruption, and Aberrant Segregation of Chromosomes

**DOI:** 10.3390/ijms241210003

**Published:** 2023-06-11

**Authors:** Liga Wuri, Robert C. Burghardt, Joe A. Arosh, Charles R. Long, Sakhila K. Banu

**Affiliations:** 1Department of Veterinary Integrative Biosciences, School of Veterinary Medicine and Biomedical Sciences, Texas A&M University, College Station, TX 77843, USA; lwuri@cvm.tamu.edu (L.W.); rburghardt@cvm.tamu.edu (R.C.B.); jarosh@cvm.tamu.edu (J.A.A.); 2Department of Veterinary Physiology and Pharmacology, School of Veterinary Medicine and Biomedical Sciences, Texas A&M University, College Station, TX 77843, USA; clong@cvm.tamu.edu

**Keywords:** oocyte, oxidative stress, DNA double-strand break, hexavalent chromium, RAD51, microtubule, polar body extrusion

## Abstract

Environmental and occupational exposure to hexavalent chromium, Cr(VI), causes female reproductive failures and infertility. Cr(VI) is used in more than 50 industries and is a group A carcinogen, mutagenic and teratogenic, and a male and female reproductive toxicant. Our previous findings indicate that Cr(VI) causes follicular atresia, trophoblast cell apoptosis, and mitochondrial dysfunction in metaphase II (MII) oocytes. However, the integrated molecular mechanism of Cr(VI)-induced oocyte defects is not understood. The current study investigates the mechanism of Cr(VI) in causing meiotic disruption of MII oocytes, leading to oocyte incompetence in superovulated rats. Postnatal day (PND) 22 rats were treated with potassium dichromate (1 and 5 ppm) in drinking water from PND 22–29 and superovulated. MII oocytes were analyzed by immunofluorescence, and images were captured by confocal microscopy and quantified by Image-Pro Plus software, Version 10.0.5. Our data showed that Cr(VI) increased microtubule misalignment (~9 fold), led to missegregation of chromosomes and bulged and folded actin caps, increased oxidative DNA (~3 fold) and protein (~9–12 fold) damage, and increased DNA double-strand breaks (~5–10 fold) and DNA repair protein RAD51 (~3–6 fold). Cr(VI) also induced incomplete cytokinesis and delayed polar body extrusion. Our study indicates that exposure to environmentally relevant doses of Cr(VI) caused severe DNA damage, distorted oocyte cytoskeletal proteins, and caused oxidative DNA and protein damage, resulting in developmental arrest in MII oocytes.

## 1. Introduction

Infertility affects approximately 186 million people, representing 8 to 12% of reproductive-aged couples worldwide [1]. Females are born with a finite supply of oocytes, and it is critical that healthy oocytes are maintained throughout reproductive life to ensure fertility, successful ovulation, fertilization, and embryonic development [2]. Various external factors, including exposure to endocrine disrupting chemicals (EDCs) such as heavy metals, plasticizers, pesticides, ionizing radiation, and chemotherapeutic agents, have adverse effects on genomic integrity and the health of oocytes [3,4]. Studies in mouse models, mammalian cells, and other model organisms have provided important insights into the pathogenesis of human diseases related to defects in genome maintenance [5]. Many factors involved in genome maintenance play crucial roles during embryonic development, as well as later in life. Mutations in genes involved in DNA replication, repair, and checkpoint pathways underlie several genetically inherited disorders [5]. Therefore, rapid detection and repair of damaged DNA determine good oocyte quality. Failure in proper and timely DNA damage repair leads to defective oocytes with aneuploidy, embryo lethality, or birth defects [5]. Several studies have used microarrays to profile RNA expression in germinal vesical (GV) oocytes, metaphase II (MII) stage oocytes, and embryos in mice and humans [2]. DNA double-strand breaks (DSBs) are considered to be the most toxic of the DNA damage types, as they can cause genomic rearrangements and structural changes such as deletions, translocations, and fusions in the DNA, which typically result in loss of the cellular function or viability of oocytes [3].

Interestingly, components from all DNA repair pathways, including direct lesion reversal, base excision repair (BER), mismatch repair (MMR), nucleotide excision repair (NER), homologous recombination (HR), and non-homologous end-joining repair (NHEJ), are represented in mouse, monkey, and human MII oocytes and embryos [2]. BER is the primary DNA repair pathway that corrects base lesions arising from oxidative, alkylation, deamination, and depurination/depyrimidination damage [6]. Phosphorylation of histone H2AX on serine 139 (γ-H2AX) is the most recognized assay to measure DNA damage and offers an initial indication for cell death, with each γ-H2AX focus representing an individual DNA DSB [7]. Oocytes exposed to EDCs such as glyphosate and methoxychlor increased γ-H2AX foci, compromising DNA integrity and inducing cytotoxicity [8,9].

Reactive oxygen species (ROS), such as free radicals and oxidants, react with different components of DNA and modify the base [2]. The imbalance between ROS formation and enzymatic and non-enzymatic antioxidants (AOXs) leads to various diseases, cancer, and male and female infertility [10]. Exposure to EDCs, ionizing radiation, and smoking leads to oxidative stress [11]. Excessive oxidative stress results in oxidative DNA damage, and 8-hydroxy-2-deoxyguanosine (8-OHdG) is a well-known biomarker of oxidative DNA damage [12]. An increase in 8-OHdG may reflect oxidative DNA damage and can indicate a decline in the DNA repair rate [13]. Aging increases 8-OHdG production in the ovary of mice and is associated with infertility [14]. Even though ROS is involved in female infertility [15], the mechanism of ROS in damaging oocyte DNA and microtubule disruption is an understudied area of research. A recent study in women undergoing in vitro fertilization (IVF) demonstrated that elevated levels of the AOX catalase in the follicular fluid favored successful fertilization and live birth and had a positive IVF outcome [16]. Our previous study in rats reported that Cr(VI) induced follicular atresia by downregulating AOXs GPx1, SOD-1 and -2, catalase, PRDX3, and TXN-2 in the antral follicles, which was mitigated by supplementing resveratrol to Cr(VI)-exposed rats [17].

Superoxide radical in the presence of cellular nitric oxide forms peroxynitrite (ONOO^−^), which in turn nitrates tyrosine residues forming nitrotyrosine (NTY), leading to changes in protein structure and function [18]. Oxidative damage to protein is characterized by structural modification of side chains by ROS, including oxidation of sulfhydryl groups, oxidative adducts on amino acid residues near metal-binding sites, cross-linking, unfolding, and protein fragmentation. Direct ROS attack on the amino acid side chains of proline, arginine, lysine, and threonine results in protein carbonyl formation, which eventually changes the tertiary structure of a protein and alters protein function [19]. Our previous study showed that Cr accumulation in the human placenta is associated with increased levels of NTY [20]. However, the role of NTY in altering the protein structure of the oocytes is largely unknown.

Cr(VI) is one of eight metals in the top fifty priority list for toxic substances made by the Agency for Toxic Substances and Disease Registry [21]. Increased usage and improper disposal of chromium waste into the environment increases Cr contamination in the water, air, and soil [22]. A recent report by Environment Texas reported the discharge of Cr waste into Calaveras Lake, San Antonio, Texas, posing a potential risk to public health [23]. Leather tanning industries contribute to the extreme Cr contamination in Southeast Asia, exposing workers to various health hazards [24]. Environmental and health advocates indicated that the Cr contamination in Calaveras Lake threatens the health of anglers, boaters, and people who eat fish from the lake [23]. Chromate workers from Painesville, Ohio, reported having a reduction in lung cancer incidence from 1940 to 1965 when Cr content in the air was reduced from 270 mg/m^3^ to 39 mg/m^3^ [25]. In addition, the environmental exposure of women living in Willits, California, to Cr(VI) caused adverse health effects, abortions, pregnancy complications, and infertility in the mothers (direct exposure) and their daughters (in utero exposure) [26]. Even though the genotoxic effects of Cr(VI) in lung cancer have been well established, Cr(VI)-induced genotoxicity in the oocytes is unknown. Our recent study indicated that exposure to Cr(VI) in young rats caused infertility by disrupting cytoskeletal machinery and mitochondrial function, resulting in dysmorphic oocytes [27].

The United States Environmental Protection Agency (USEPA) has established a non-enforceable Maximum Contaminant Level Goal (MCLG) and an enforceable Maximum Contaminant Level (MCL) for total chromium in drinking water systems. The USEPA MCLG and MCL are identical, set at 100 ppb for total chromium [28]. As of July 2011, the Office of Environmental Health Hazard Assessment of the California EPA established a Public Health Goal (PHG) specific to Cr(VI) at 100 ppb [28]. However, groundwater from Midland, Texas, contains 5.28 ppm Cr [29]. A large population in the US is potentially consuming drinking water with high Cr(VI) levels; therefore, Cr(VI)-induced infertility in females needs to be thoroughly studied in the light of human Cr(VI) exposure levels.

Oocytes undergo meiotic maturation before fertilization. In mammals, the process is initiated with the breakdown of the nuclear envelope and the formation of a bipolar microtubule spindle in the center of the egg that undergoes a series of dynamic reconfigurations to capture, sort, and align the chromosomes to the equatorial plate [30]. Under the active guidance and control of cytoplasmic actin, the spindle is positioned to the cortex, resulting in the first polar body (PB1) extrusion. Failures in the dynamics of cytoskeletal proteins, mainly F-actin, frequently lead to chromosome misdistributions and abnormal embryo development, causing infertility, miscarriages, and congenital diseases in humans. The increased risk for aneuploidy is associated with the unstable nature of meiotic spindles and aberrant chromosome attachments occurring during spindle assembly [31]. These critical features of meiosis in mammalian oocytes and how they are modified by EDCs are unclear. The role of actin within the spindle during oocyte maturation remains poorly understood. The close association of F-actin and microtubules at the spindle implies a functional interdependence between the two systems for proper spindle function [30]. Disturbance in the orchestration of cytoskeletal machinery will result in aneuploidy.

Aneuploidy is a significant factor contributing to implantation failure and early miscarriage in human embryos. More than half of human embryos are affected by aneuploidy, resulting in miscarriage or birth defects [32]. X chromosome aneuploidy predisposes humans to autoimmune diseases, cancer, primary biliary cirrhosis, congenital defects, and genetic disorders such as X chromosome monosomy, Turner’s syndrome, Down syndrome, and progeroid pathologies [33]. Exposure to EDCs is associated with oocyte aneuploidy due to chromosome missegregation and microtubule defects [9,34,35,36]. EDCs such as phthalates and BPA induce aneuploidy during oocyte meiosis [37,38]. However, the molecular mechanism of Cr(VI) in causing chromosome missegregation during meiosis is not clearly understood. Therefore, the current study hypothesizes that exposure to environmentally relevant doses of Cr(VI) accelerates oxidative stress leading to DNA double-strand breaks and abnormal F-actin dynamics causing defective microtubule alignment, rendering abnormal and poor quality oocytes in superovulated rats. 

## 2. Results

### 2.1. Exposure to Cr(VI) Distorted Microtubule Structure and Chromosome Arrangement in Metaphase II Oocytes

Healthy oocytes from the MII phase carry symmetrical and barrel-shaped microtubules with proper chromosome alignment in the metaphase plate. Compromised chromosome alignment and disrupted microtubule architecture are detrimental to the oocytes, increasing the risk of abortion or birth defects in children. Strikingly, current data show that exposure to 1 ppm or 5 ppm Cr(VI) results in severely misaligned chromosomes with highly disrupted microtubules in various shapes and patterns (Figure 1 and Figure 2). While oocytes from the control group exhibited healthy bipolar oocytes, the Cr(VI)-exposed oocytes exhibited various abnormal phenotypes in terms of microtubule shapes and chromosome distribution. Exposure to Cr(VI) distorted microtubule orientation and chromosome distribution, presenting microtubules in various abnormal patterns (non-polar, uneven polar, distorted, bulged bipolar (Figure 1), unfocused, tripolar, bulged bipolar, and asymmetric multipolar with misaligned chromosomes (Figure 2)). Cr(VI) significantly (*p* < 0.05) increased the percentage of abnormal microtubules (Figure 3A), and the abnormal microtubule structure was accompanied by thickened and highly folded peripheral oocyte F-actin with abundant expression (Figure 3B).

### 2.2. Exposure to Cr(VI) Delayed Polar Body Extrusion and Resulted in Incomplete Cytokinesis in MII Oocytes

Our data show that Cr(VI) increased F-actin accumulation in the MII oocytes, resulting in folded and thickened actin caps (Figure 4F,N) accompanied by incomplete cytokinesis and a delay in PB extrusion (Figure 4H,P). 

### 2.3. Exposure to Cr(VI) Increased Oxidative DNA Damage in the MII Oocytes

8-OHdG is a biomarker for oxidative DNA damage [39]. As shown in Figure 5, exposure to Cr(VI) at a dose of 1 or 5 ppm increased 8-OHdG levels in the MII oocytes compared to the control. An accelerated rate of 8-OHdG excision and migration towards the periphery of the oocytes (extracellular release) under Cr(VI) exposure suggests upregulation of the base excision repair pathway (Figure 5A–J).

### 2.4. Exposure to Cr(VI) Increased Oxidative Protein Damage in the MII Oocytes

Nitrotyrosine is a biomarker for oxidative protein damage [14]. To examine whether Cr(VI) causes oxidative damage to protein in the MII oocytes, we determined NTY levels. As depicted in Figure 6, exposure to Cr(VI) at 1 and 5 ppm doses increased NTY levels in the MII oocytes compared to the control (48.15 ± 3.7 and 39 ± 2.8 vs. 4.6 ± 0.2). However, 1 ppm Cr(VI) caused maximum expression of NTY compared to 5 ppm (Figure 6M). 

### 2.5. Exposure to Cr(VI) Increased DNA Double-Strand Breaks in MII Oocytes 

We determined the DNA double-strand break marker γH2AX in MII oocytes. Cr(VI) at 1ppm significantly increased γH2AX expression compared to the control (IOD value: 7.69 ± 0.48 vs. 1.38 ± 0.25; *p* > 0.05). Cr(VI) at 5 ppm (12.91 ± 0.47) significantly increased γH2AX expression compared to Cr(VI) at 1 ppm (7.69 ± 0.48) and the control (1.38 ± 0.25; *p* > 0.05) (Figure 7). 

### 2.6. Exposure to Cr(VI) Increased RAD51 Expression in MII Oocytes

RAD51, a multifunctional protein, plays a central role in DSB repair and replication fork processing [40]. Since Cr(VI) increased DNA DSBs, we examined RAD51 expression in the MII oocytes. Our data show that exposure to 1 ppm (11.0 ± 3.1) and 5 ppm (24.1 ± 3.2) Cr(VI) increased the expression of RAD51 in the MII oocytes compared to the control (4.1 ± 3.0) (Figure 8).

## 3. Discussion

Female fertility has declined over the past half-century [41]. Several factors contribute to the decline, including genetic factors, lifestyle factors, and exposure to EDCs, i.e., natural or synthetic exogenous compounds that interfere with the physiology of normal endocrine-regulated events such as reproduction and growth [42]. A recent study reported thirteen EDCs in human follicular fluid samples, with the highest concentrations belonging to nonylphenol and the insecticide mirex [43]. Previous findings showed a clear linear correlation between EDC concentration and a reduced maturation and fertilization rate for polychlorinated biphenyls, polybrominated diphenyl ethers, dichlorodiphenyltrichloroethane, nonylphenol, and mirex. Studies from animal models revealed adverse effects of chronic EDC exposure during the resumption of meiosis I or II, thus limiting the process of oocyte maturation and fertilization [44,45]. Most EDCs cause DNA damage and alter gene transcription [3], leading to various gynecological diseases and infertility. 

With growing industrialization and exposure to EDCs, women experience challenges with infertility, resulting in a dependence on Assisted Reproductive Techniques (ARTs) [4]. According to the Center for Disease Control, there were 326,468 ART cycles performed at 449 reporting clinics in the United States during 2020, resulting in 75,023 live births [46]. Overcoming poor oocyte quality is the major limiting factor determining the IVF success rate [47]. Oocytes play a critical role in correcting DNA damage by preventing apoptosis and inhibiting the transmission of genetic mutations to offspring, thus preserving fertility [3,4]. 

Cr(VI) rapidly enters cells by mimicking anions through transporters such as sulfate ion transporters [48]. Cellular reductants such as enzymatic and non-enzymatic AOXs convert Cr(VI) into Cr(III), where Cr(III) forms Cr(III)-DNA adducts, causing mutations and DNA strand breaks [48]. Our recent study demonstrated that Cr(VI) caused infertility in rats by increasing F-actin and disrupting mitochondrial function in MII oocytes [27]. Our previous study reported that gestational exposure to Cr(VI) caused premature ovarian failure (POF) in rats by altering *xpnpep-2*, a POF marker gene [49]. Environmental and occupational exposures to Cr(VI) increased infertility in women [21]. However, how Cr(VI) affects the integrity of DNA or disrupts spindle assembly by altering microtubules is unknown. A recent study reported that a high dose of Cr(VI) (50 ppm) given to mice for 21 days induced microtubule disruption, which was inhibited by quercetin [36]. Our study, for the first time, reveals that environmentally relevant doses of Cr(VI) (1 and 5 ppm) caused severe oocyte deficiencies, including abnormal and distorted microtubules, incomplete polar body extrusion, and increased numbers of misaligned and disintegrated chromosomes with severe spindle abnormality due to increased actin deposition. A study from mouse oocytes revealed that actin depletion drastically affected chromosome alignment during metaphase II [50]. In the current study, abnormal chromosome attachments to the spindle led to the disarrangement of chromosomes, as well as a loss of bipolarity due to the abnormal deposition and folding of F-actin. There is increasing evidence that actin in the spindle assembly participates in spindle migration and positioning and protects oocytes from chromosome segregation errors leading to aneuploidy [30]. Actin is an integral component of the meiotic machinery that closely interacts with microtubules during all major events of oocyte maturation (from the time point of spindle assembly until PB extrusion and metaphase arrest) [30]. In healthy oocytes, PB extrusion through polar relaxation occurs via specific weakening of the cortical patch, which is caused by local depletion of actomyosin contractility [51]. Interestingly, Cr(VI) increased F-actin abundance and the thickness of the cortical patch, resulting in delayed or incomplete cytokinesis and PB extrusion.

We further explored DNA strand breaks and oxidative stress to understand the mechanism of chromosome missegregation and disintegration. We established ROS as one of the primary pathways for Cr(VI) to cause apoptosis of the oocytes [52] and placental trophoblasts [53], whereas antioxidants such as resveratrol [17], edaravone [52], and ascorbic acid [54] mitigated Cr(VI)-induced follicular atresia in rats. Indeed, increased ROS levels in follicular fluid have been associated with poor oocyte and embryo qualities and low pregnancy rates in women [55]. The most important oxygen-free radical causing damage to basic biomolecules (proteins, membrane lipids, and DNA) is the hydroxyl radical (HO•). Cr(VI) produces HO• by the Fenton reaction, and the HO• attacks DNA strands when it is produced adjacent to cellular and mitochondrial DNA, leading to the generation of various oxidation products [56]. The interaction of HO• with the nucleobases of the DNA strand, such as guanine, leads to the formation of C8-hydroxyguanine (8-OHGua) or its nucleoside form 8-OHdG. The 8-OHdG undergoes keto–enol tautomerism, which favors the oxidized product 8-oxo-7,8-dihydro-2’-deoxyguanosine (8-oxodG). In the scientific literature, 8-OHdG and 8-oxodG are used for the same compound [39]. Occupational exposure to EDCs such as PAH, benzene, styrene, and inorganic arsenic increases urinary 8-oxodG in workers [57]. 

Cases of miscarriage and stillbirths due to heavy metal exposure continue to rise in developing nations. Occupational exposure is often cited as a risk factor for female fertility, as well as for early pregnancy loss and pre-term delivery [58]. Non-occupational (environmental) exposure to Cr(VI) in Willits, California, caused detrimental reproductive effects in human females and their infants [26]. In the European Union, many Cr(VI) compounds are classified as reprotoxic substances [59]. An association between pregnancy loss and parental exposure to stainless steel welding was reported [59]. In vitro studies showed that Cr(VI) caused cell cycle arrest in granulosa cells by altering cell cycle regulatory proteins, with potential intervention by vitamin C [60], activation of mitochondria-mediated intrinsic apoptotic pathways, and p53 activation [61]. In addition, several experimental studies indicate that Cr(VI) is a potent reproductive toxicant causing follicular atresia, pregnancy failure, premature ovarian failure, abnormal placental development due to increased oxidative stress, and impaired antioxidant activity [59]. Whether heavy metals alter DNA in the oocytes or embryos of women due to occupational exposure is unknown. The current data show that Cr(VI) increases 8-OHdG in the MII oocytes of superovulated rats. An accelerated rate of 8-OHdG excision and migration towards the periphery of the oocytes (extracellular release) under Cr(VI) exposure suggests upregulation of the BER pathway. Our study supports a recent study where exposure to H_2_O_2_ increased excision and migration of 8-OHdG towards the periphery of MII oocytes [62]. The oocyte-derived BER removes and replaces a single damaged nucleotide by targeting the damaged base. Our study suggests BER as one of the preferred DNA repair mechanisms that oocytes adapt against Cr(VI)-induced genotoxicity.

Consistent with the increase in 8-OHdG, Cr(VI) increased NTY in the MII oocytes, a hallmark of ONOO^−^, thus indicating interaction between NO and O_2_^−^. ONOO^−^ promotes nitration of the tyrosine residues of cellular proteins, depletes lipid-soluble antioxidants, and initiates lipid peroxidation [63]. Enhanced O_2_^−^ and increased production of ONOO^−^ lead to poor oocyte quality in women with endometriosis [63]. ONOO^−^ can cross biological membranes and oxidize iron–sulfur centers of proteins, eventually affecting oocyte quality [64]. Direct ROS attack on the amino acid side chains changes a protein’s tertiary structure and results in protein function alterations. ROS deteriorates postovulatory MII oocyte quality and integrity [65]. Our previous study showed a positive association between increased NTY accumulation and elevated Cr burden in human term placenta [20]. Collectively, our data show that Cr(VI) increased oxidative stress in DNA and proteins, which adversely affected the integrity of genome and cytoskeletal proteins (microtubules and F-actin) in MII oocytes and resulted in severe oocyte deterioration.

DNA damage accumulation increases chromosomal fragmentation and affects meiosis, spindle assembly, and mitochondrial distribution in the oocyte, ultimately affecting embryo development [3]. DSBs are the most toxic of the DNA damage types, which increases deletions, translocations, and fusions in the DNA, resulting in loss of oocyte viability [3]. Pb chromate induced DNA DSBs in human lung cells [66]. DNA DSBs represent a major concern for the maintenance of oocyte genomic integrity. If left unrepaired, they can result in meiotic arrest and apoptosis, and if repaired incorrectly, they can potentially lead to delayed or abnormal chromosomal segregation. Aneuploidy, caused by the aberrant segregation of chromosomes, can lead to miscarriage or birth defects such as trisomy 21 (Down syndrome), trisomy 18 (Edwards syndrome), and trisomy 13 (Patau syndrome) [67]. Exposure to EDCs, aging, and lifestyle factors can contribute to aneuploidy of the oocytes [68,69]. Aging has also been associated with decreased expression of the key genes involved in the DNA repair of DSBs in human and mouse oocytes [3]. Therefore, we determined DNA DSBs in the MII oocytes in response to Cr(VI). Our data show that Cr(VI) increased DNA DSBs, evidenced by an increased amount of the DNA DSB marker γ-H2AX. 

The homologous recombination DNA DSB repair pathway is the predominant pathway in oocytes from the GV to MII stage, while NHEJ becomes the predominant DNA repair mechanism post fertilization. DNA repair protein RAD51 homolog 1 (RAD51), a highly conserved protein among species, plays a central role in HR repair of DNA breaks. HR depends on the formation of a RAD51 recombinase filament that facilitates strand invasion [3]. Microinjection of recombinant RAD51 before irradiation prevented irradiation-induced DNA damage in both bovine and mouse oocytes [40]. To determine if Cr(VI) alters the DNA repair mechanism, we investigated RAD51. Cr(VI) activated the expression of RAD51 compared to control oocytes. Thus, the current study suggests that Cr(VI) increased DNA DSBs, with DNA DSBs in turn inducing the activation of DNA damage repair machinery, namely the expression of RAD51. Previous studies show that exposure to EDCs such as BPA [70] and zearalenone [71] causes DNA strand breaks and elevated expression of RAD51, supporting the current finding. 

Taken together, as depicted in the schematic diagram in Figure 9, our data show that Cr(VI) (i) increased oxidative stress in DNA, causing base modification and DNA double-strand breaks (DSBs); (ii) increased oxidative protein damage and disrupted oocyte cytoskeletal protein F-actin and microtubules; and (iii) activated abnormal accumulation and folding of the F-actin filament, resulting in thickened actin caps, incomplete cytokinesis, and delayed PB1 extrusion. (iv) All of these cytoskeletal disruptions led to distorted microtubules and misaligned chromosomes, which might lead to aneuploidy, embryo lethality, infertility, or birth defects.

## 4. Materials and Methods

### 4.1. Chemicals 

The chemicals used in these studies were purchased from Sigma Chemical Company (St. Louis, MO, USA), ABCAM Inc. (Boston, MA, USA), Fisher Scientific Company LLC (Houston, TX, USA), or Life Technologies Cooperation (Carlsbad, CA, USA) unless stated otherwise.

### 4.2. Animals and Treatments

Sprague Dawley (SD) rats were purchased from Charles River Laboratories (Houston, TX, USA), maintained in AAALAC-approved animal facilities with a 12 h light/12 h dark cycle at 23–25 °C, and fed with Teklad 4% mouse/rat diet and water ad libitum. Animal use protocols were performed following the National Institute of Health Guidelines for the Care and Use of Laboratory Animals and with standards established by Guiding Principles in the Use of Animals in Toxicology and specific guidelines and standards of the Society for the Study of Reproduction approved by the Animal Care and Use Committee of Texas A&M University. Postnatal day (PND) 22 female rats were divided into three groups: control (n = 20), Cr(VI)-1 ppm (n = 20), and Cr(VI)-5 ppm (n = 20) for the metaphase II (MII) oocytes collection. Control rats received regular drinking water, and Cr(VI)-treated groups received 1.0 ppm or 5.0 ppm potassium dichromate in drinking water from PND 22 to 28. 

Rationale for choosing Cr(VI) doses and age of Cr(VI) exposure: As elaborated in the Introduction, the drinking water Cr level in Midland, TX, USA, is 5.28 ppm. Based on the behavioral and physiological indicators, the developing rats from PND 22 to 30 represent the juvenile group. We chose PND 22–30 for Cr(VI) exposure and superovulation since PND 30 is the peripubertal age and the optimal age in rats for maximum egg retrieval.

### 4.3. Superovulation and Oocyte Collection

Superovulation and oocytes collection was performed as we recently described [27]. Briefly, Sprague Dawley rats were exposed to Cr(VI) for seven days prior to intraperitoneal (i.p) injection with 10 IU of pregnant mare serum gonadotropin (cat. no. RP1782721000, Bio Vendor, Brno, Czech Republic). After 48 h, rats were injected with 10 IU human chorionic gonadotropin (hCG) (cat. no. C1063, Sigma-Aldrich, St. Louis, MO, USA), and 14–15 h later, the rats were euthanized and oocytes collected for various analyses.

### 4.4. Immunocytochemistry of γ-H2AX, RAD 51, 8-OHdG, Nitrotyrosine, and Microtubules

For immunofluorescence staining, cumulus–oocyte complexes were pulled out from the ampulla region of the superovulated rat oviduct into 1 mg/mL hyaluronidase to remove the surrounding cumulus cells from the oocytes. Cumulus cell-free oocytes were fixed in 4% paraformaldehyde for 30 min and transferred to the permeabilization solution (containing 0.3% polyvinylpyrrolidone (PVP) + 0.1% Tween-20 + 0.01% Triton X-100) for 20 min [72]. After fixation, oocytes were incubated with a blocking buffer (Invitrogen™, Waltham, MA, USA) for 1 h. For γ-H2AX staining, oocytes from each group were incubated in the recombinant Alexa Fluor^®^ 647 anti-gamma H2AX (phospho S139) antibody (cat. no. ab195189; 1:200 dilution) for 1 h at room temperature (RT). Oocytes were next stained for microtubules through incubation with the mouse monoclonal anti-α-tubulin–FITC antibody (clone DM1A, purified from hybridoma cell culture, diluted 1:200 with blocking buffer, Sigma, St. Louise, MO, USA; catalog no. F2168) at RT for 1 h. After three washes with 0.3% PVP washing media, the oocytes from each group were stained for F-actin through incubation with 1 μg/mL of phalloidin-tetramethylrhodamine (rhodamine phalloidin; cat. no. R415; Invitrogen, Waltham, MA, USA) at RT for 30 min. After 3 washes in the 0.3% PVP, oocytes were transferred into a small drop of prolong antifade mounting medium containing 4,6-diamidino-2-phenylindole (DAPI) on a microscope slide and covered with a coverslip. For RAD 51, 8-OHdG, and nitrotyrosine (NTY) staining, oocytes were incubated with primary antibodies (anti-Rad51 antibody, cat. no. ab133534; anti-8-OHdG antibody, cat. no. ab 48508; and anti-NTY antibody, cat. no. ab125106) in the blocking buffer overnight, washed three times in the washing buffer, transferred to the corresponding secondary antibodies at RT for 1 h, and then stained for microtubules and F-actin. After three washes in the 0.3% PVP washing media, the oocytes were transferred into a small drop of prolong antifade mounting medium containing DAPI on a glass slide and covered with a coverslip. The confocal images were captured with a Zeiss LSM 780 microscope. The intensity of staining for 8-OHdG, NTY, γ-H2AX, and RAD51 was quantified using Image-Pro Plus 6.3 software, Version 10.0.5 (Media Cybernetics, Inc., Bethesda, MD, USA) and expressed as Integrated Optical Density (IOD) according to the manufacturer’s instructions.

### 4.5. Image Quantification

The expression levels of proteins in immunofluorescent images were quantified by Image Pro-Plus software, Version 10.0.5 as reported previously [49,52,73,74,75]. The confocal images were captured with a Zeiss LSM 780 microscope. The intensity of staining for each protein was quantified using Image-Pro Plus^®^ 10.0.5 image processing and analysis software according to the manufacturer’s instructions (Media Cybernetics, Inc., Bethesda, MD). The detailed methods for quantification are given in the instruction guide: “The Image-Pro Plus: The proven solution for image analysis”. In brief, the integrated optical intensity (IOD) of immunostaining was quantified under the RGB mode using the Image-Pro Plus software selection tools. Numerical data were expressed as least square mean ± SEM. This technique is more quantitative than conventional blind scoring systems.

### 4.6. Statistical Analysis

One-way analysis of variance (ANOVA) followed by Tukey post-test pair-wise comparison was used to detect the statistical significances from the mean of all the quantitative data analyzed for the control, Cr 1 ppm, and Cr 5 ppm groups using the statistical software SigmaPlot 14.5 (Systat Software Inc., Chicago, IL, USA). *p* < 0.05 was considered statistically significant. The Shapiro–Wilk test was used to test the normality of data (*p* = 0.622).

## Figures and Tables

**Figure 1 ijms-24-10003-f001:**
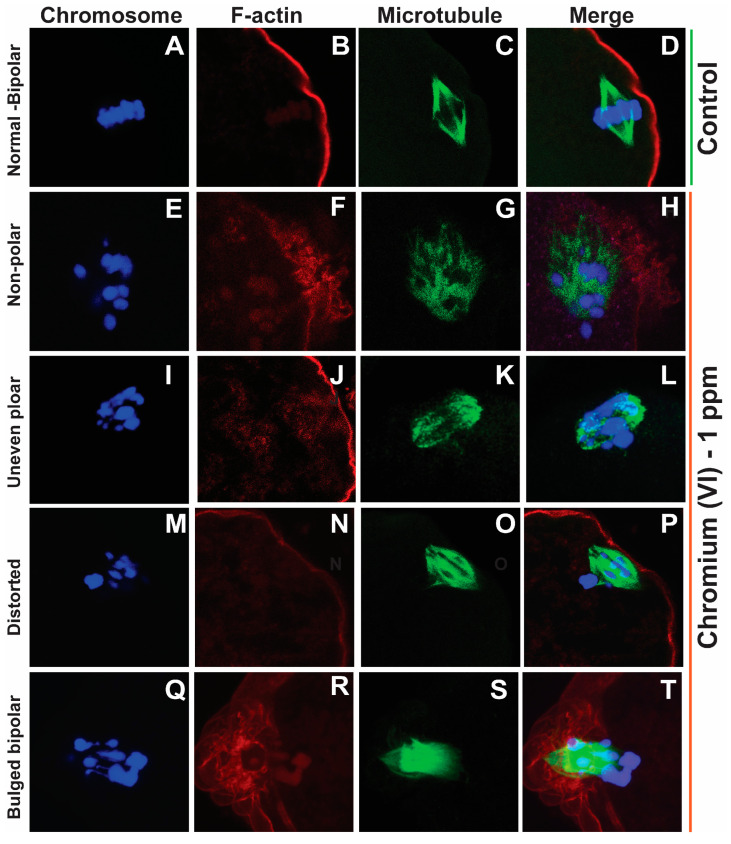
Effects of 1 ppm Cr(VI) on microtubule structure and chromosome alignment in rat metaphase II oocytes. Prepubertal rats were exposed to 1 ppm potassium dichromate through drinking water from PND 22 to 29 and superovulated. Immunofluorescence was performed in the MII oocytes. All images were captured with a 40×/1.4 NA Plan-Apochromat lens, and the width of each field is 35 µm. Representative images are shown. Chromosomes are shown in blue (**A**,**E**,**I**,**M**,**Q**), F-actin is shown in red (**B**,**F**,**J**,**N**,**R**), microtubules are shown in green (**C**,**G**,**K**,**O**,**S**), and merged images are also shown (**D**,**H**,**L**,**P**,**T**). (**A**–**D**) Control: bipolar spindle with normal chromosome alignment. Cr(VI) 1 ppm: (**E**–**H**) Non-polar spindle: chromosomes are misaligned and are scattered, with the spindle lacking polarity. (**I**–**L**) Uneven polar spindle: spindle poles are unfocused, and chromosomes are not evenly aligned. (**M**–**P**) Distorted spindle: spindle with broad poles and dispersed chromosomes. (**Q**–**T**) Bulged bipolar: bipolar spindle bulged in the middle with uneven chromosome alignment.

**Figure 2 ijms-24-10003-f002:**
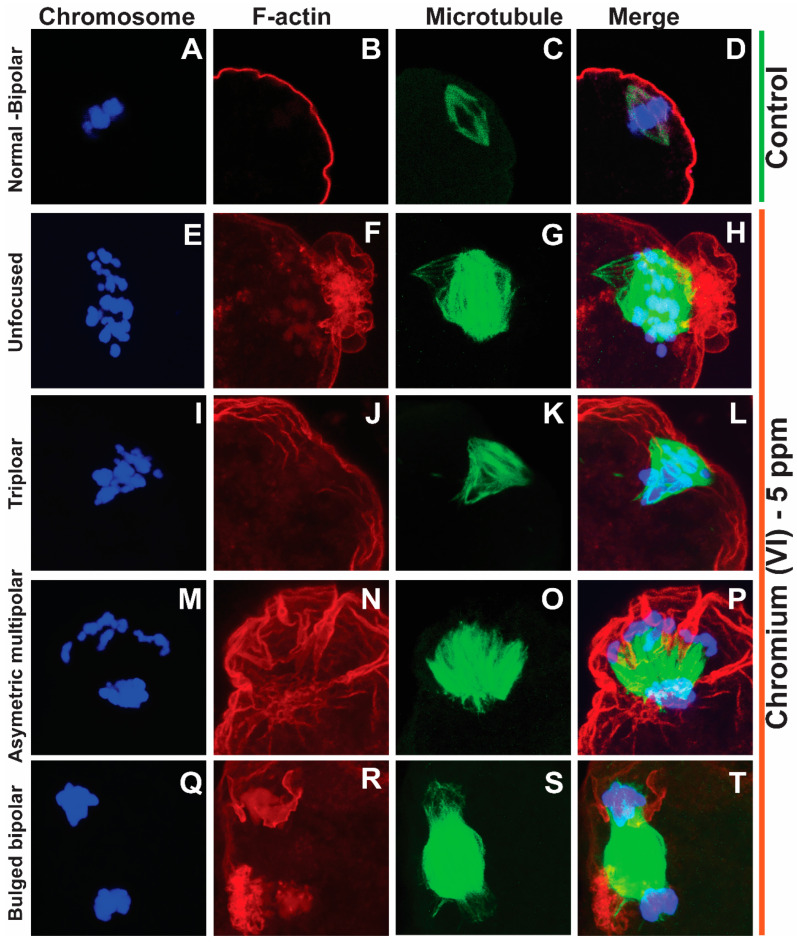
Effects of 5 ppm Cr(VI) on microtubule structure and chromosome alignment in rat metaphase II oocytes. Prepubertal rats were exposed to 1 ppm potassium dichromate through drinking water from PND 22 to 29 and superovulated. Immunofluorescence was performed in the MII oocytes. All images were captured with a 40×/1.4 NA Plan-Apochromat lens, and the width of each field is 35 µm. Representative images are shown. Chromosomes are shown in blue (**A**,**E**,**I**,**M**,**Q**), F-actin is shown in red (**B**,**F**,**J**,**N**,**R**), microtubules are shown in green (**C**,**G**,**K**,**O**,**S**), and merged images are also shown (**D**,**H**,**L**,**P**,**T**). (**A**–**D**) Control: bipolar spindle with normal chromosome alignment. Cr(VI) 5 ppm: (**E**–**H**) Unfocused chromosome and abnormal spindle with several misaligned chromosomes. (**I**–**L**) Tripolar spindle with unfocused poles and chromosomes aligned in three directions. (**M**–**P**) Asymmetric multipolar maple leaf-shaped spindle with extremely broad poles and dispersed chromosomes. (**Q**–**T**) The bipolar spindle bulged in the middle. F-actin (**H**,**L**,**P**,**T**) shows an extreme folding pattern, tightly surrounding the chromosomes where the microtubules are tangled within F-actin.

**Figure 3 ijms-24-10003-f003:**
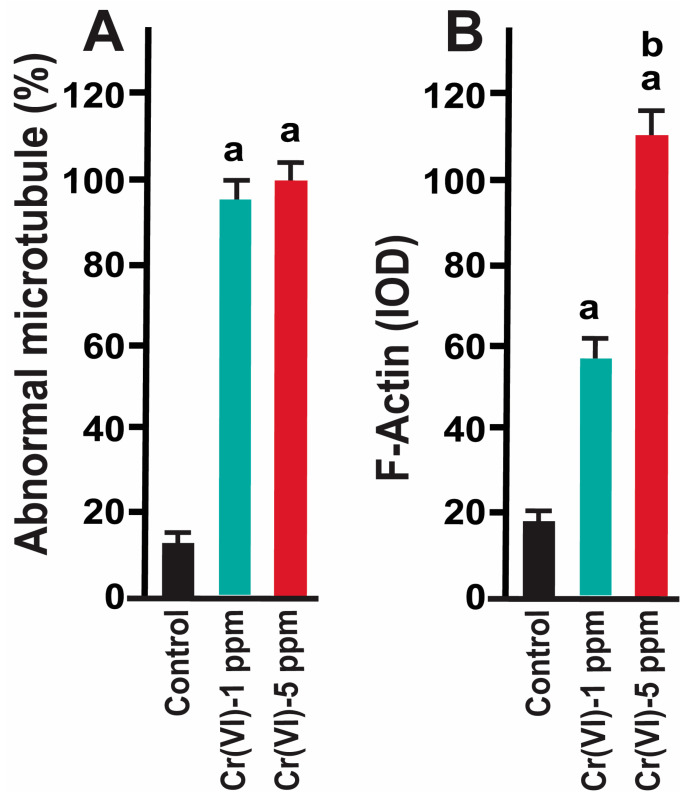
Effects of Cr(VI) exposure on abnormal microtubule structure and F-actin expression. Prepubertal rats were exposed to 1 ppm or 5 ppm potassium dichromate through drinking water from PND 22 to 29 and superovulated. Immunofluorescence was performed in the MII oocytes. (**A**) The number of oocytes with dispersed chromosomes and abnormal microtubules was counted and expressed as a percentage. Each value is mean ± SEM of 100 oocytes from 10 rats. (**B**) Expression of F-actin was determined by immunofluorescence. Images were captured by confocal microscopy and quantified using Image-Pro Plus software, Version 10.0.5 (Media Cybernetics Inc.). a: control vs. Cr(VI) 1 ppm or 5 ppm; b: Cr(VI) 1 ppm vs. 5 ppm; *p* < 0.05.

**Figure 4 ijms-24-10003-f004:**
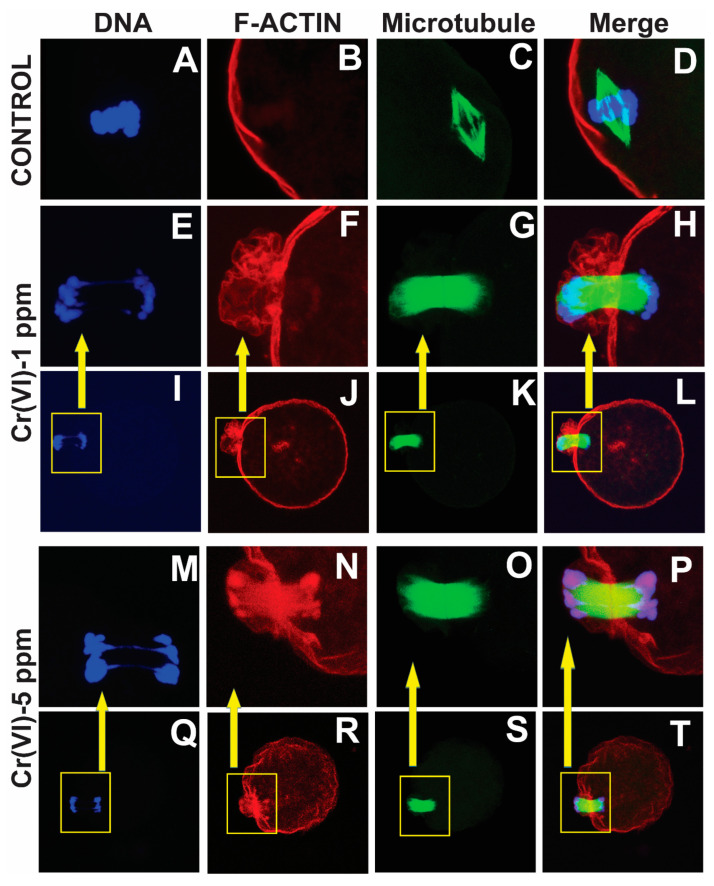
Effects of Cr(VI) exposure on polar body extrusion. Prepubertal rats were exposed to 1 or 5 ppm potassium dichromate through drinking water from PND 22 to 29 and superovulated. Immunofluorescence was performed in the MII oocytes, images were captured by confocal microscopy, and representative images are shown. All images were captured with a 40×/1.4 NA Plan-Apochromat lens. The width of each field is 35 µm (**A**–**H**,**M**–**P**) or 100 µm (**I**–**L**,**Q**–**T**). Boxed inserts in (**I**–**L**,**Q**–**T**) is 35 µm. Cr(VI) caused the thickening of the actin cap, resulting in incomplete cytokinesis and delayed PB1 extrusion. Control (**A**–**D**) and Cr(VI) 1 ppm (**E**–**L**) and 5 ppm (**M**–**T**).

**Figure 5 ijms-24-10003-f005:**
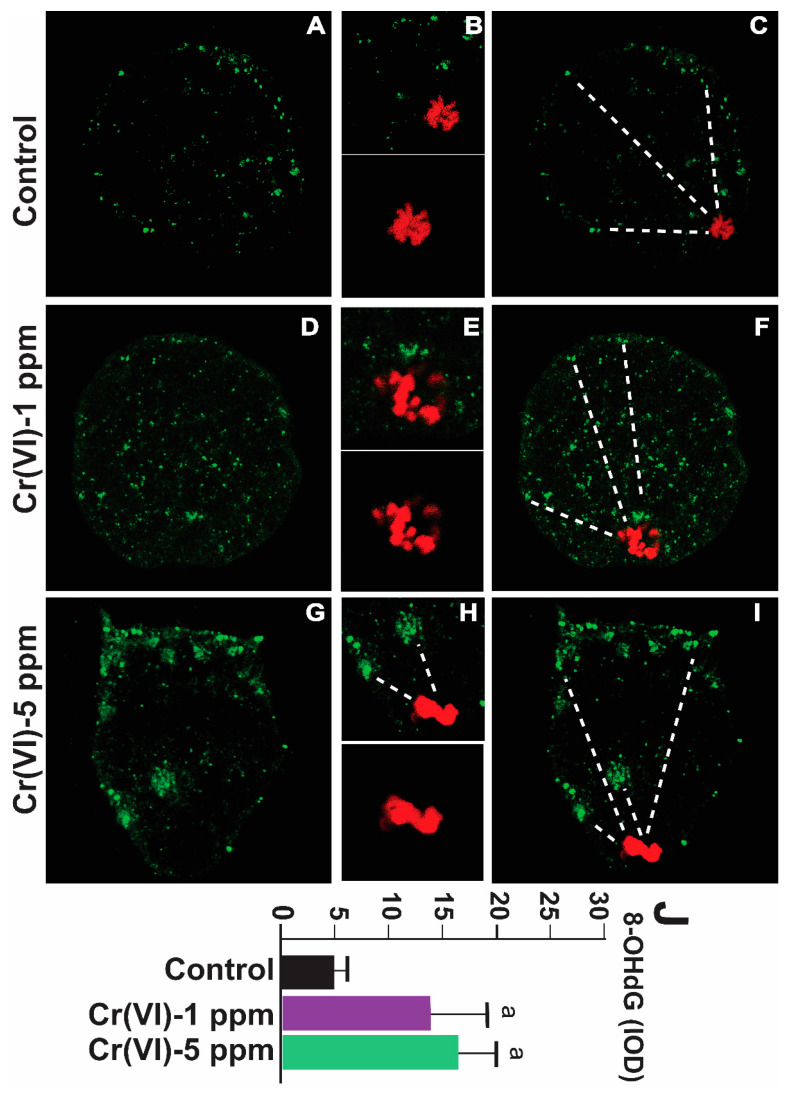
Effects of Cr(VI) exposure on oxidative DNA damage (8-OHdG) in MII oocytes. Prepubertal rats were exposed to 1 or 5 ppm potassium dichromate through drinking water from PND 22 to 29 and superovulated. 8-OHdG, the biomarker of oxidative DNA damage, was determined by immunofluorescence in the MII oocytes, and images were captured by confocal microscopy. All images were captured with a 40×/1.4 NA Plan-Apochromat lens. The width of each field is 50 µm (**A**,**C**,**D**,**F**,**G**,**I**) or 15 µm (**B**,**E**,**H**; top panel) or 10 µm (**B**,**E**,**H**; bottom panel). Cr(VI) increased 8-OHdG expression. Representative images of the control (**A**–**C**), 1 ppm Cr(VI) (**D**–**F**), and 5 ppm Cr(VI) (**G**–**I**) groups are shown. The histogram (**J**) represents the intensity of staining (expressed as Integrated Optical Density (IOD)). Each value is mean ± SEM of ~24 oocytes from six rats (*p* < 0.05). a: control vs. Cr(VI) 1 ppm or 5 ppm. The red color represents chromosomes, and the green color represents 8-OHdG. Broken lines show 8-OHdG excision and migration towards the periphery of the oocytes (extracellular release).

**Figure 6 ijms-24-10003-f006:**
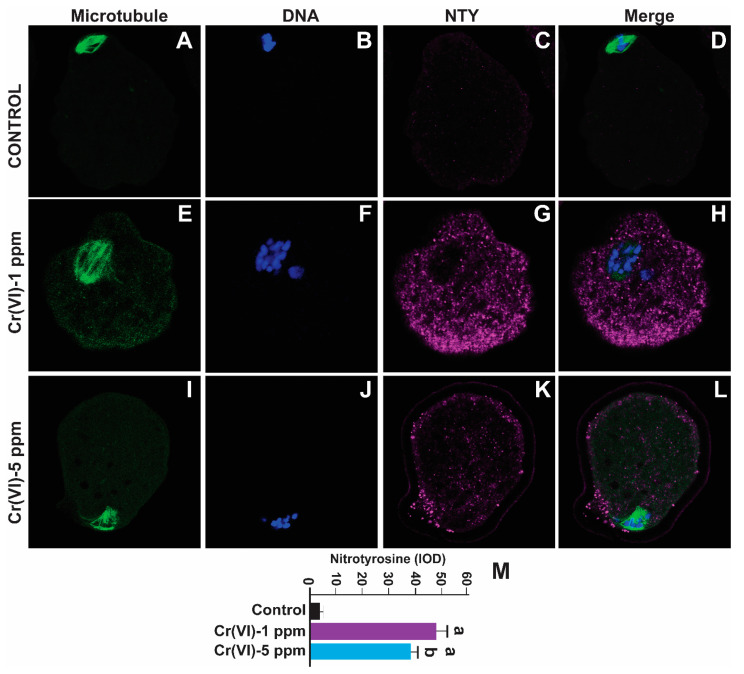
Effects of Cr(VI) exposure on oxidative protein damage in MII oocytes. Prepubertal rats were exposed to 1 or 5 ppm potassium dichromate through drinking water from PND 22 to 29 and superovulated. Nitrotyrosine, a biomarker of oxidative protein damage, was determined by immunofluorescence in the MII oocytes. Images were captured by confocal microscopy; all images were captured with a 40×/1.4 NA Plan-Apochromat lens and the width of each field is 70 µm. Images quantified using Image-Pro Plus software, Version 10.0.5 (Media Cybernetics Inc.). Cr(VI) increased NTY expression. Representative images of the control (**A**–**D**), 1 ppm Cr(VI) (**E**–**H**), and 5 ppm Cr(VI) (**I**–**L**) groups are shown. The histogram (**M**) represents the intensity of staining (expressed as Integrated Optical Density (IOD)). Each value is mean ± SEM of ~24 oocytes from six rats (*p* < 0.05). a: control vs. Cr(VI) 1 ppm or 5 ppm.

**Figure 7 ijms-24-10003-f007:**
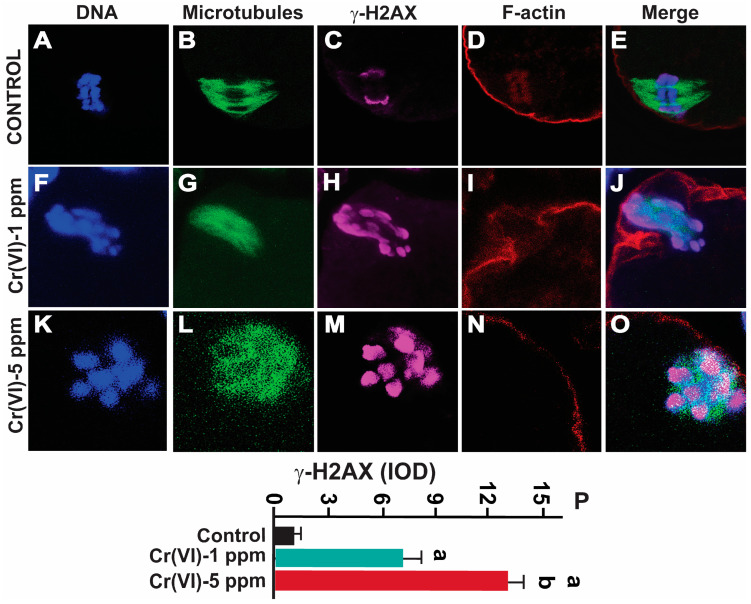
Effects of Cr(VI) exposure on DNA double-strand breaks. Prepubertal rats were exposed to 1 or 5 ppm potassium dichromate through drinking water from PND 22 to 29 and superovulated. γ-H2AX, the DNA DSB marker, was determined by immunofluorescence. All confocal images were captured with a 40×/1.4 NA Plan-Apochromat lens and the width of each field is 25 µm. Images were quantified using Image-Pro Plus software, Version 10.0.5 (Media Cybernetics Inc.). Representative images of the control (**A**–**E**), 1 ppm Cr(VI) (**F**–**J**), and 5 ppm Cr(VI) (**K**–**O**) groups are shown. The histogram (**P**) represents the intensity of staining (expressed as Integrated Optical Density (IOD)). Each value is mean ± SEM of ~24 oocytes from six rats (*p* < 0.05). a: control vs. Cr(VI) 1 ppm or 5 ppm; b: Cr(VI) 1 ppm vs. 5 ppm.

**Figure 8 ijms-24-10003-f008:**
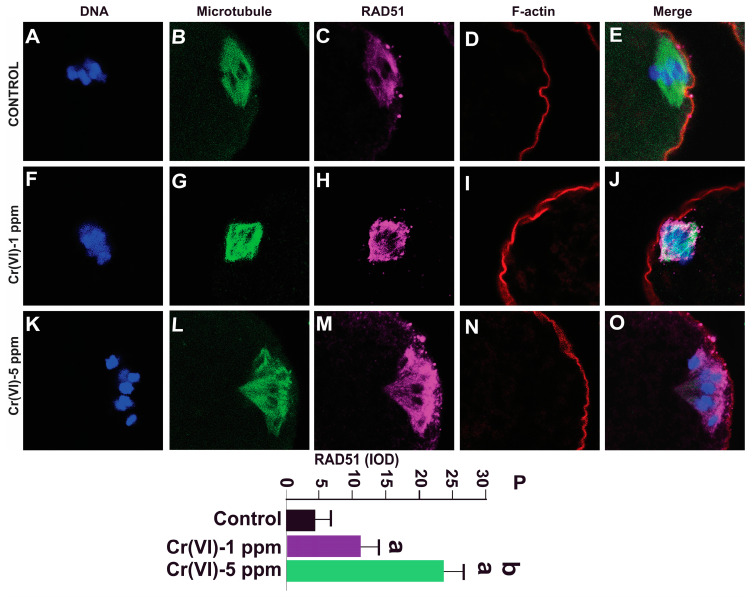
Effects of Cr(VI) exposure on DNA damage repair protein RAD51. Prepubertal rats were exposed to 1 or 5 ppm potassium dichromate through drinking water from PND 22 to 29 and superovulated. RAD51, a DNA damage repair protein, was determined by immunofluorescence. All confocal images were captured with a 40×/1.4 NA Plan-Apochromat lens and the width of each field is 35 µm. Images were quantified using Image-Pro Plus software, Version 10.0.5 (Media Cybernetics Inc.). Representative images of the control (**A**–**E**), 1 ppm Cr(VI) (**F**–**J**), and 5 ppm Cr(VI) (**K**–**O**) groups are shown. The histogram (**P**) represents the intensity of staining (expressed as Integrated Optical Density (IOD)). Each value is mean ± SEM of ~24 oocytes from six rats (*p* < 0.05). a: control vs. Cr(VI) 1 ppm or 5 ppm; b: Cr(VI) 1 ppm vs. 5 ppm.

**Figure 9 ijms-24-10003-f009:**
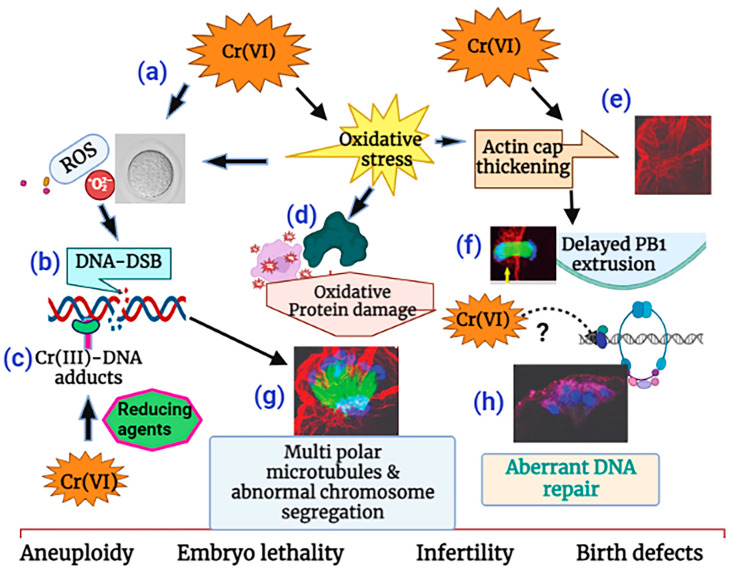
Schematic diagram of the mechanism of Cr(VI)-induced meiotic disruption of MII oocytes. (**a**) Cr(VI) increased oxidative stress in DNA, (**b**) causing base modification and DNA double-strand breaks (DSB). Cr(VI), after being reduced to Cr(III), forms (**c**) Cr(III)-DNA adducts. Cr(VI) increased (**d**) oxidative protein damage and (**e**) F-actin thickening, leading to (**f**) delayed polar body extrusion. (**g**) Cr(VI)-induced cytoskeletal disruptions led to distorted microtubules and misaligned chromosomes and was accompanied by (**h**) aberrant DNA repair, which might lead to aneuploidy, embryo lethality, and infertility. This illustration is created using www.biorender.com.

## Data Availability

Data is contained within the article.

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
