# Peer review of "Hexavalent Chromium Disrupts Oocyte Development in Rats by Elevating Oxidative Stress, DNA Double-Strand Breaks, Microtubule Disruption, and Aberrant Segregation of Chromosomes"

_ijms, 2023, doi:10.3390/ijms241210003_

Round 1

Reviewer 1 Report

The manuscript “Hexavalent chromium induces infertility in rats by elevating oxidative stress, DNA double-strand breaks, microtubule disruption, and aberrant segregation of chromosomes in the metaphase-II oocytes” by Liga Wuri, Robert C Burghardt, Joe A Arosh, Charles R Long and Sakhila K. Banu is presenting interesting new data about the negative effects of hexavalent chromium in the female fertility. However, there are several issues to be clarified.

Introduction

1)     Line 54: Here, only the abbreviation is necessary.

2)     Line 89: Only abbreviation is necessary here.

Materials and methods

Animals and treatments

3)     It´s hard to follow the amount of Cr(VI) used and correlate with the maximum contaminant level. Please, use one unit only (or ppm or ug/L).

4)     The paragraph from lines 154 until 165 would be better placed in the introduction.

Statistical analysis

5)     You repeted the last phrase.

Figures

6)     Figure 1 is misplaced.

7)     Would be better to reorganise figures 1 and 2 in only one (Figure 1 = fig.1 + fig. 2).

8)     Figure 3 could be smaller.

9)     The legend should be understood without the help of material and methods. Please, add the necessary information in all the legends.

10)  Figure 5. What is green, red? Please, add these info.

11)  Figure 7. Please, turn the figure 90 degrees (as all the others, specially, Figure 8). It´s nicer to compare with all the other figures.

Results

12)  In this section, only your results should be presented. No explanation, discussion. Please, review lines 225-227; 258-259; 276-277; 287-288; 302-304.

Discussion

13)  Please, add a reference for the information “leading to various gynaecological diseases and infertility (line 343-344). 

Author Response

REVIEWER-1

The manuscript “Hexavalent chromium induces infertility in rats by elevating oxidative stress, DNA double-strand breaks, microtubule disruption, and aberrant segregation of chromosomes in the metaphase-II oocytes” by Liga Wuri, Robert C Burghardt, Joe A Arosh, Charles R Long and Sakhila K. Banu is presenting interesting new data about the negative effects of hexavalent chromium in the female fertility. However, there are several issues to be clarified.

The authors gratefully acknowledge the positive comments and constructive critiques by the reviewer, which helped improve the quality of the manuscript.

Introduction

Comment 1:   Line 54: Here, only the abbreviation is necessary.

Response: In line 54, we kept the abbreviation as per the reviewer’s comment.

Comment 2: Line 89: Only abbreviation is necessary here. We followed the reviewer’s suggestion.

Response: we kept the abbreviation as per the reviewer’s comment.

Materials and methods, Animals and treatments

Comment 3: It´s hard to follow the amount of Cr(VI) used and correlate with the maximum contaminant level. Please, use one unit only (or ppm or ug/L).

Response: We apologize for the oversight. We kept ppm and removed ug/L.

Comment 4: The paragraph from lines 154 until 165 would be better placed in the introduction.

Response: The authors thank the reviewer for the suggestion. We have moved the paragraph to the introduction.

Statistical analysis

Comment 5: You repeated the last phrase.-

Response: Removed in the revised manuscript.

Figures

Comment 6: Figure 1 is misplaced.

Response: Thank you. We have correctly placed it in the revised manuscript.

Comment 7: Would be better to reorganise figures 1 and 2 in only one (Figure 1 = fig.1 + fig. 2).

Response: The authors appreciate the reviewer for the suggestion. However, the two doses of Cr(VI), 1 and 5 ppm, had unique effects on the microtubules. F-actin activation and chromosomal missegregation is severe with 5 ppm compared to 1 ppm Cr(VI). To show the uniqueness and/or the similarities of the different doses of Cr(VI) and also to keep the nice layout of the figures, we prefer to keep the two figures separate. If the reviewer or editor approves, we want to change it to Figure 1A (1 ppm Cr) and 1B (5 ppm Cr).

Comment 8: Figure 3 could be smaller.

Response: We have reduced the size. The authors thank the reviewer for the suggestion.

Comment 9: Figure The legend should be understood without the help of material and methods. Please, add the necessary information in all the legends.

Response: We revised the legend throughout as per the reviewer’s suggestion.

Comment 10: Figure 5. What is green, red? Please, add this info.

Response: Green represents 8-OHdG, and red represents DNA. We added info in the figure-5 legend.

Comment 11: Figure 7. Please, turn the figure 90 degrees (as all the others, especially, Figure 8). It´s nicer to compare with all the other figures.

Response: The authors thank the reviewer for catching the mistake. It’s a great suggestion. We have rearranged the figure according to the reviewer’s suggestion.

Results

Comment 12:  In this section, only your results should be presented. No explanation, or discussion. Please, review lines 225-227; 258-259; 276-277; 287-288; 302-304.

 Response: We have removed the specified lines in the revised manuscript.

Discussion

Comment 13: Please, add a reference for the information “leading to various gynecological diseases and infertility (lines 343-344). 

Response: Most EDCs cause DNA damage and alter gene transcription leading to various gynecological diseases and infertility. Winship et al., 2018 are the ref for the entire text. We moved the ref to the end of the sentence.

Reviewer 2 Report

The manuscript is well planned and written, but some clarifications are required.  They are as follows:

1. Unlcear whether immunocytochemistry studies of oxidative stress and protein damage markers (8-hydroxy-2-deoxyguanosine and nitrotyrosine) and microtubules are convincing to evaluate the competence of the oocytes. The application of biomarkers analyzed by RT-qPCR and microarray analysis (for examples, BMP15, OSF, and Cx43) would have significantly enhanced the scientific interest of the manuscript.

2. Were all the metaphase II (MII) oocytes collected after superovulation  (L171)? What about in vitro maturation (IMV)? Should provide explanations.

3. Infertility was not examined in this study. The Reviewer suggests to rephase the title and in the text where its appears. As matter of fact endocrine disrupting chemicals (EDC) impair oocyte developmental competence.

Some minor corrections.

Author Response

AUTHORS' RESPONSES TO REVIEWER-2

Comment 1: The manuscript is well planned and written, but some clarifications are required.  They are as follows:

Response: The authors appreciate the reviewer for the positive feedback on the manuscript and the valuable critique in improving the manuscript.

Comment 2: Unlcear whether immunocytochemistry studies of oxidative stress and protein damage markers (8-hydroxy-2-deoxyguanosine and nitrotyrosine) and microtubules are convincing to evaluate the competence of the oocytes. The application of biomarkers analyzed by RT-qPCR and microarray analysis (for examples, BMP15, OSF, and Cx43) would have significantly enhanced the scientific interest of the manuscript.

 Response: The authors appreciate the reviewer for the important input. Out goal is to delineate the mechanisms of Cr(VI) in causing meiotic errors in the metaphase-II oocytes (which leads to aneuploidy). 8-OHdG and nitrotyrosine are well-known markers for oxidative DNA and protein damage, respectively. We highly appreciate the reviewer’s suggestion to use RT-qPCR and microarray analysis of the oocytes exposed to Cr(VI). Our lab is currently working on RNA-seq analysis of oocytes exposed to Cr(VI).

Comment 3: Were all the metaphase II (MII) oocytes collected after superovulation (L171)? What about in vitro maturation (IMV)? Should provide explanations.

Response: Yes, all the oocytes were collected after superovulation. We did not perform any vitro maturation in the current study.

Comment 4: Infertility was not examined in this study. The Reviewer suggests to rephrase the title and in the text where its appears. As a matter of fact endocrine-disrupting chemicals (EDC) impair oocyte developmental competence.

Response: We thank the reviewer for the important input. As per the reviewer’s comment, we removed the word “infertility” and replaced it with “oocyte developmental competence”. The new title of the revised manuscript is “Hexavalent chromium disrupts oocyte development in rats by elevating oxidative stress, DNA double-strand breaks, microtubule disruption, and aberrant segregation of chromosomes”.

Reviewer 3 Report

The MS merite major revision. 

 - In the abstract can you please add mean-SD and the p-value?

- In the introduction can you please add data regarding infertility around the world?  

- Can you please specify why did you choose such doses of Cr and the time of administration?

- I suggest adding details regarding the software of  Image ProPlus software. 

- Please revise the reference style throughout the manuscript.

- Have the authors tested their data normality?

- Please add the catalog numbers for the applied kits

- please add to the histograms the statistical significativity of the letter a B...

- please add value to the results part

- the discussion is well written, I suggest adding more articles regarding the effect of Cr in vitro in vivo and clinical study. 

Author Response

AUTHORS' RESPONSES TO REVIEWER-3

The authors are very grateful for the reviewer’s appreciation of the manuscript (Discussion) and very insightful criticism. We apologize for the oversight, and we address every critique by the reviewer as follows:

 Comment 1: In the abstract can you please add mean-SD and the p-value?

Response: We added the details in the abstract. Please see lines 21-22.

Comment 2: In the introduction can you please add data regarding infertility around the world?  -

Response: Data on global infertility is added. Please see lines 32-33.

Comment 3: Can you please specify why did you choose such doses of Cr and the time of administration?

Response: We have included the rationale for choosing the doses of Cr(VI) in the revised manuscript (please see lines 165-169). We chose juvenile rats (22-30 days old) as this is the best age in rats to retrieve maximum oocytes through superovulation.

Comment 4: I suggest adding details regarding the software of Image ProPlus software. 

Response: We have added details in the revised manuscript. Please see lines 205-215.

Comment 5: Please revise the reference style throughout the manuscript.

Response: We have used endnote to format the IJMS reference style. We have checked the references.

Comment 6: Have the authors tested their data normality?

Response: We used the Shapiro-Wilk test as our numerical means of assessing normality.

Comment 7: Please add the catalog numbers for the applied kits

Response: We have added catalog numbers.

Comment 8: please add to the histograms the statistical significativity of the letter a B...

Response: We apologize for the oversight. We have added the details in the figure legends.

Comment 9: please add value to the results part

Response: We have added numerical values of the results and p-values.

Comment 10: The discussion is well written; I suggest adding more articles regarding the effect of Cr in vitro in vivo and clinical studies. 

Response: We have included more literature/background in the discussion section regarding the effect of Cr in vitro in vivo and clinical study. 

Round 2

Reviewer 3 Report

I would like to thank authors because they have addressed all my requests. However, for the abstract I meant that the authors add the numerical values and the p-value as expressed by Mean ± SEM, So can you please correct and delete the sentence of ''The values 20 were expressed as mean ± SEM....., p≤0.05.''

Author Response

Following is a comment from Reviewer 3, and please see our response:   Comment from reviewer 3:  I would like to thank the authors because they have addressed all my requests. However, for the abstract, I meant that the authors added the numerical values and the p-value as expressed by Mean ± SEM, So can you please correct and delete the sentence "The values 20 were expressed as mean ± SEM....., p≤0.05."  

Response: The authors thank the reviewer for the comment. We deleted the sentence as the reviewer suggested. Per the reviewer's suggestion, we have already added the numerical values in the results section. Therefore, we indicated approximate fold changes in the abstract instead of repeating the numerical values, as we usually add numerical values under the results section.  
